# Implications of Spatially Constrained Bipennate Topology on Fluidic Artificial Muscle Bundle Actuation

Emily Duan * and Matthew Bryant

Department of Mechanical and Aerospace Engineering, North Carolina State University, Raleigh, NC 27695, USA; mbryant@ncsu.edu
* Correspondence: eduan@ncsu.edu

**Abstract:** In this paper, we investigate the design of pennate topology fluidic artificial muscle bundles under spatial constraints. Soft fluidic actuators are of great interest to roboticists and engineers, due to their potential for inherent compliance and safe human–robot interaction. McKibben fluidic artificial muscles are an especially attractive type of soft fluidic actuator, due to their high force-to-weight ratio, inherent flexibility, inexpensive construction, and muscle-like force-contraction behavior. The examination of natural muscles has shown that those with pennate fiber topology can achieve higher output force per geometric cross-sectional area. Yet, this is not universally true for fluidic artificial muscle bundles, because the contraction and rotation behavior of individual actuator units (fibers) are both key factors contributing to situations where bipennate muscle topologies are advantageous, as compared to parallel muscle topologies. This paper analytically explores the implications of pennation angle on pennate fluidic artificial muscle bundle performance with spatial bounds. A method for muscle bundle parameterization as a function of desired bundle spatial envelope dimensions has been developed. An analysis of actuation performance metrics for bipennate and parallel topologies shows that bipennate artificial muscle bundles can be designed to amplify the muscle contraction, output force, stiffness, or work output capacity, as compared to a parallel bundle with the same envelope dimensions. In addition to quantifying the performance trade space associated with different pennate topologies, analyzing bundles with different fiber boundary conditions reveals how bipennate fluidic artificial muscle bundles can be designed for extensile motion and negative stiffness behaviors. This study, therefore, enables tailoring the muscle bundle parameters for custom compliant actuation applications.

**Keywords:** biomimetic; pennate topology; soft actuators; fluidic artificial muscles; muscle topology

## 1. Introduction

Actuators are vital to enabling mechatronic systems to interface with the physical world. Thus, roboticists and engineers have placed a great deal of attention on the design of actuators, and many have drawn inspiration from the distinctive attributes of biological muscles to equip actuators with safe human–robot interaction capabilities. Neuromuscular physiology and anatomy studies have shown that single biological muscle tissue is composed of multiple motor units [1]. Each motor unit consists of hundred to thousands of muscle fibers that can be arranged in a variety of topologies. Analogously, hierarchical actuators use multi-unit architectures to augment the total actuator performance and increase actuator functionality [2,3]. The practicality of this hierarchical actuation strategy has been demonstrated on cellular piezoelectric actuators [1], whiffletree actuators [2], series-parallel elastic actuators [4], and bio-inspired orderly recruitment actuator bundles [5]. This muscle-inspired hierarchy has led to the development of linear bundle actuators capable of mimicking orderly recruitment, and thus improving efficiency by activating only the smallest required actuator unit for a given task [5–7]. Recent studies have explored parallel and pennate topologies of bundle actuators, as well as the influence of initial braid angle on

contraction, extension, and twisting behaviors [4,8–10]. The parallel topology configures the longitudinal axis of individual actuator 'fibers' parallel to the bundle actuator line of motion, while the pennate topology orients individual actuator fibers at an angle to the bundle line of motion. These topologies are bio-inspired: human skeletal muscle tissues exhibit parallel (e.g., biceps brachii muscle), asymmetric unipennate (e.g., extensor digitorum or posterior forearm muscle), and symmetric bipennate (e.g., rectus femoris in the quadriceps muscle group) architectures [11]. Advantages of the pennate topology in biological muscles and the effects of fiber pennation angle on contraction speed, damping from impact disturbances, and aging have been emphasized in numerous studies [8,9,12,13]. Some studies explored arranging pneumatic fiber-reinforced actuators in pennate-inspired networks to illustrate how an inverted pennation arrangement can further amplify net displacement, as compared to a more conventional pennate configuration [14]. The coupling between the muscle force and displacement to the fiber force and displacement results in the ability to passively alter the effective gear ratio (i.e., the effective relationship between muscle fiber input and macro muscle outputs) [8,9,13]. This passive control of effective gear ratio has led to the categorization of pennate actuator bundles as variable stiffness actuators, which are especially attractive for their potential in energy storage. Furthermore, appropriately controlled variable stiffness may allow for safer human–robot interaction [15].

McKibben fluidic artificial muscles are particularly suitable for pennate bundle actuators, due to their muscle-like actuation behavior, inexpensive construction, inherent flexibility, and high force-to-weight ratio. The dense, helical braid sleeve around the soft elastomer allows it to generate forces of hundreds to thousands of Newtons. While previous studies have established models for the force generation, contraction, and stiffness mechanics of pennate McKibben artificial muscle bundle actuators [8,13,16], there has yet to be a study to identify the relationships between pennation topology and actuator output characteristics for a given actuator envelope.

This paper explores the parameterization, design, and analysis of a bio-inspired pennate topology fluidic artificial muscle (FAM) bundle actuator under spatial envelope constraints. The primary contributions of this paper are to establish the sensitivity of FAM bundle actuation performance characteristics, including force, contraction, work output, and stiffness to pennation angle and fiber boundary condition, when bundle envelope dimensions are held constant. This paper introduces a more closely bio-inspired boundary condition, where the FAMs in the bundle are in contact, as compared to the more commonly studied pinned boundary condition. The results reveal that bipennate topologies, when appropriately designed, can achieve force, contraction, work, or stiffness performances, exceeding that of a parallel topology muscle bundle with equivalent envelope dimensions, but that tradeoffs exist between fiber boundary conditions. If the muscle bundle is configured with contacting fibers, competing effects of fiber radius and length give rise to extensile motions and negative stiffnesses for some designs. The remainder of this paper is organized as follows: Section 2 presents the system formulation with a method for muscle bundle parameterization for a prescribed bundle spatial envelope for fiber bounding conditions, Section 3 discusses and examines the effects of boundary conditions and pennation angle on muscle bundle performance, and Section 4 enumerates the conclusions of the paper.

## 2. System Formulation

### 2.1. Muscle Topologies

A bundle actuator consists of multiple McKibben fluidic artificial muscles (FAMs), where individual FAMs are referred to as "fibers", and the complete bundle actuator is referred to as a "muscle". Fibers in a pennate topology are arranged at an angle to the direction of muscle bundle motion. The fibers in a pennate muscle are subject to axial contraction, radial expansion, and rotation upon actuation. A special case of fibers with zero pennation angle is also recognized as the parallel topology. The critical difference in this zero pennation angle configuration is that the fibers will only contract axially and expand radially, but not rotate. Each muscle topology in this analysis is considered as a

single-layer, two-dimensional array of fibers. All fibers have the same initial braid angle $\alpha_i$. The fibers in the pennate topology are assumed to be symmetrically oriented in a bipennate arrangement. Figure 1 is a visual representation of the two muscle topologies under consideration. Figure 1a illustrates the special case zero pennation configuration or parallel fiber topology, and Figure 1b shows the bipennate fiber topology. Similar schematics of this pennate arrangement originate as far back as 1664 from functional biomechanics models [17]. The fibers are represented as red cylinders.

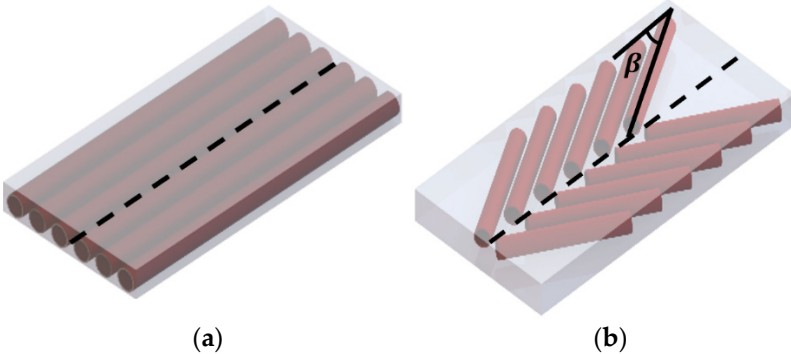

| (**a**) | (**b**) |

**Figure 1.** (**a**) Parallel topology; (**b**) Bipennate topology. Three-dimensional visual representation of muscle topologies considered. The shaded region indicates the bounding envelope. The black dashed centerline is the muscle axis of motion. The $\beta$ shown in (**b**) indicates the pennation angle or angle at which the fiber is oriented with respect to the muscle line of motion.

### 2.2. Design Case and Fiber Boundary Conditions

The goal of this study is to understand the effects of configuring the bundle with different initial fiber pennation angles, while maintaining uniform envelope dimensions of the bundle. We take the bundle envelope, shown as the shaded region in Figure 1, as the smallest fixed prismatic rectangular volume that all portions of the bundle remain within throughout actuation. Therefore, for each initial pennation angle, the parameters including fiber number, initial length, and initial radius must be determined, such that the configuration remains within the prescribed envelope during actuation. From among the set of configurations that would satisfy this criterion for each initial pennation angle, we consider a design that sets the initial fiber radius, length, and number, such that the initial total internal volume of the fibers within the envelope is as large as possible.

We also consider two sets of fiber boundary conditions:

- The *'pinned boundary condition'*, illustrated in Figure 2a, defines that one end of each fiber is pinned to the rigid frame, while the other end is pinned to a central spine. Because the pin joints are fixed to rigid bodies, an initial clearance must be provided between the fibers, such that the fibers do not interfere with one another during contraction and the resulting radial expansion. We consider the case where this clearance is set, such that the fibers contact one another at exactly one contraction condition, and never interfere over the full stroke of the muscle.

- The *'fiber contact boundary condition'* defines that contact between fibers is maintained during actuation. Fibers are assumed to remain cylindrical and are constrained to remain in frictionless, tangential contact, such that they can freely slide relative to one another, but cannot be separated. Physically, this could be implemented by pinning one end of the outer or top-most pair of fibers to a rigid external frame, while the remaining fiber ends on the rigid frame can slide freely. The opposite ends of the fibers are connected to their respective symmetric counterparts to complete the bipennate arrangement. Figure 2b illustrates a visual representation of fibers under this boundary condition. This fiber contact boundary condition represents an idealization of the function of the connective tissue in biological muscles that surrounds individual muscle fibers and enables fibers to slide relative to one another, while holding the

fibers together transversally. This connective tissue, or endomysium, is deformable to adapt to volumetric changes during muscle fiber contraction, and has been shown to have a limited role in transmitting muscular force [18]. This fiber contact boundary condition thus serves as a direct analogy to biological muscle physiology.

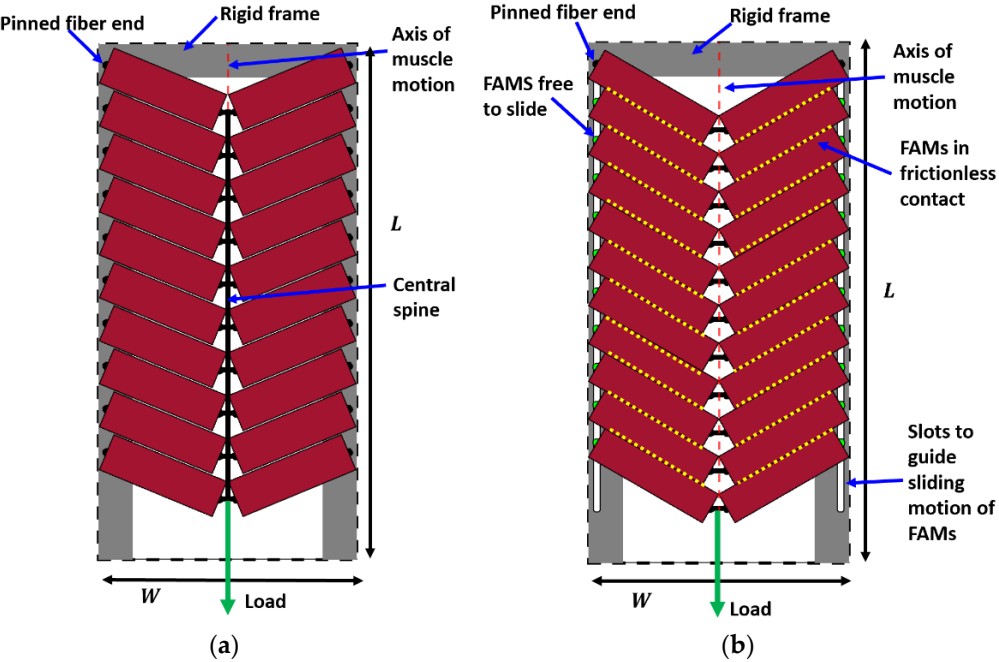

**Figure 2.** Two-dimensional visual representation of (**a**) pinned boundary condition and (**b**) fiber contact boundary condition of a bipennate muscle. The length and width of the bounding envelope are illustrated. The muscle is shown in planar view so that the depth dimension *D* is not indicated. The bounding envelope is indicated by the black dashed lines around the rigid frame. The red, dashed vertical line represents the axis of muscle motion, and the green arrow indicates the applied load. The black semicircles indicate the pinned fiber end locations on the rigid frame and central carriage. The green semicircles indicate the fiber ends to translate vertically along the rigid frame and central carriage. Symmetrical slots on the frame and central carriage in (**b**) are shown to direct the vertical translation of the FAMs. Yellow dotted lines in (**a**) indicate FAM surfaces in frictionless contact.

While the fiber contact boundary condition is more closely bio-inspired, the pinned boundary condition is mechanically simpler, because it eliminates the need for sliding degrees of freedom within the muscle.

### 2.3. Muscle Bundle Parameterization

This section details a method to determine the set of fiber parameters based on a prescribed initial pennation angle in this muscle bundle design. The initial pennation angle in combination with the fiber contraction behavior will dictate the fiber behavior during muscle contraction. The fibers in the topology can be contraction-limited, rotation-limited, or both at free contraction of the muscle. If the configuration is rotation-limited, it indicates that the fibers are not capable of reaching their full contraction potential, but will reach full rotation. Therefore, the pennation angle at free muscle contraction $\beta_{free} = 90°$, as shown in Figure 3a–c. If the configuration is contraction-limited, it indicates that the fibers are not capable of reaching full rotation, but will reach their full contraction, as illustrated in Figure 3d–f. Therefore, a contraction-limited configuration will have $\beta_{free} < 90°$ and the fiber braid angle at free contraction $\alpha_{free} = \alpha_{max}$, where $\alpha_{max}$ is the fiber braid angle corresponding to the maximum attainable fiber contraction at a given pressure. If the configuration is both contraction- and rotation-limited, it indicates that the fiber will reach full contraction and full rotation simultaneously, and thus $\alpha_{free} = \alpha_{max}$ and $\beta_{free} = 90°$.

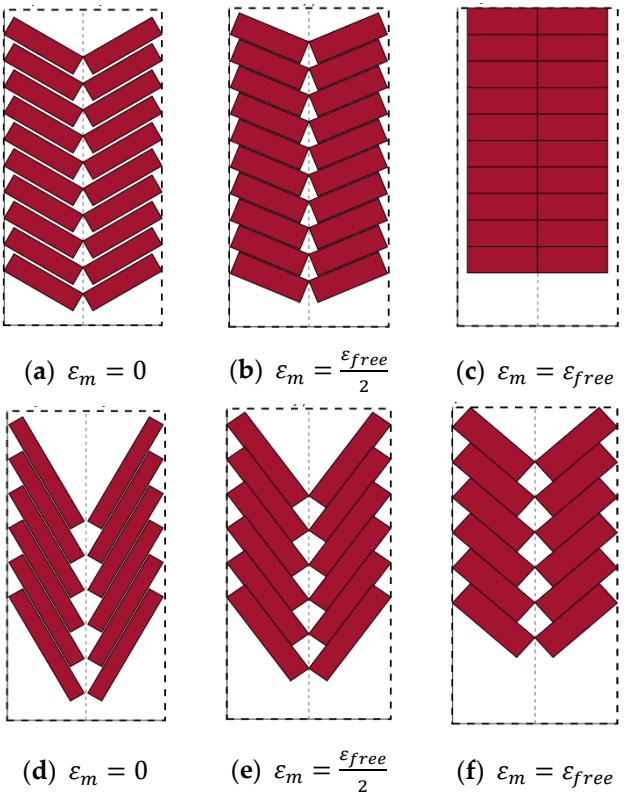

**(a)** $\varepsilon_m = 0$ **(b)** $\varepsilon_m = \dfrac{\varepsilon_{free}}{2}$ **(c)** $\varepsilon_m = \varepsilon_{free}$

**(d)** $\varepsilon_m = 0$ **(e)** $\varepsilon_m = \dfrac{\varepsilon_{free}}{2}$ **(f)** $\varepsilon_m = \varepsilon_{free}$

**Figure 3.** Two-dimensional visual representation of fiber behaviors during muscle contraction from zero muscle strain $\varepsilon_m$ to muscle strain at free contraction $\varepsilon_{free}$. Each image shows a snapshot of the fiber dimensions and orientation at a stage of muscle strain. Images in the top row (**a**–**c**) illustrate rotation-limited bipennate configuration ($\beta_i = 60°$), and the images in the bottom row (**d**–**f**) illustrate a contraction-limited configuration ($\beta_i = 60°$). The planar view of the bounding envelope is indicated by the black dashed lines.

To analyze this muscle contraction behavior, we consider the fibers as idealized McKibben actuators, such that the model assumes a cylindrical fiber geometry, and negligible bladder thickness and stiffness [19]. Muscle bundle thickness is held constant to enable an explicit relationship between fiber axial contraction and rotation for finding $\beta_{free}$ and $\alpha_{free}$.

$$l_{fi}\sin(\beta_i) = l_f \sin(\beta) \tag{1}$$

$$\frac{l_f}{l_{fi}} = \frac{\sin(\beta_i)}{\sin(\beta)} \tag{2}$$

The ideal McKibben artificial muscle presented by Tondu provides a relationship between the fiber length and braid angle

$$(\frac{l_f}{l_{fi}} = \frac{\cos(\alpha)}{\cos(\alpha_i)}) \cdot \frac{\sin(\beta_i)}{\sin(\beta)} = \frac{\cos(\alpha)}{\cos(\alpha_i)} \tag{3}$$

$$\beta = \sin^{-1}\left(\frac{\sin(\beta_i)\cos(\alpha_i)}{\cos(\alpha)}\right) \tag{4}$$

where $\beta_i$ is the initial fiber pennation angle and $\alpha_i$ is the initial braid angle. If the instantaneous braid angle $\alpha$ in (4) was substituted for the maximum possible braid angle $\alpha_{max} = \tan^{-1}\left(\sqrt{2}\right) \cong 54.7°$, the computed $\beta$ would give the pennation angle at free contraction $\beta_{free}$. If $\beta_{free} < 90°$, the fibers at $\beta_i$ are contraction-limited and the braid angle

at free contraction $\alpha_{free} = \alpha_{max}$. However, if $\beta_{free} = 90°$, $\beta_{free}$ must be substituted into $\beta$ in (4) and solve for $\alpha$ to give the braid angle at free contraction $\alpha_{free}$.

Relationships between fiber parameters and dimensions of the bounding envelope can be formulated as follows, to ensure that the fibers remain inside the bounding envelope during muscle contraction [15]:

$$(\frac{n_w}{2} - 1)\frac{2r_f}{\cos(\beta)} + 2r_f \cos(\beta) + l_f \sin(\beta) \leq \frac{W}{2} \tag{5}$$

$$(\frac{n_l}{2} - 1)\frac{2r_f}{\sin(\beta)} + 2r_f \sin(\beta) + l_f \cos(\beta) \leq L \tag{6}$$

where $n_w$ is the number of fibers that can fit along the prescribed width dimension of the bounding envelope $W$, $n_l$ is the number of fibers that can fit along the prescribed length dimension of the bounding envelope $L$, $r_f = \frac{\sin(\alpha)}{\sin(\alpha_i)}r_{f,i}$ is the instantaneous fiber radius, $l_f = \frac{\cos(\alpha)}{\cos(\alpha_i)}l_{f,i}$ is the instantaneous fiber length, and $\alpha$ is the instantaneous braid angle. Inequalities (5) and (6) are used to assess the muscle bundle length and thickness remain within the prescribed bounding envelope. These constraints are necessary to design a bipennate topology muscle bundle under spatial bounds. However, these are not sufficient conditions to determine the set of fiber parameters for this muscle bundle design. Furthermore, these conditions are not adequate for designing bipennate topology muscle bundles with pinned boundary conditions.

This study analyzes a design that sets the fully expanded fiber diameter such that the initial volume of the muscle bundle is maximized within the spatial bounds. Therefore, the fiber radius at free muscle contraction is bounded as $0 < r_{f,free} \leq \frac{D}{2}$. For this range of possible fiber radii at free muscle contraction, corresponding fiber parameters can be determined for each initial pennation angle to obtain the set of fiber parameters that gives the largest initial muscle bundle volume

$$V_m = n\pi r_{f,i}^2 l_{f,i} \tag{7}$$

for each initial pennation angle, where $n = \max(n_w, n_l)$.

For a possible $r_{f,free}$ within the bounds, the initial fiber radius $r_{f,i}$ can be derived as

$$r_{f,i} = \frac{\sin(\alpha_i)}{\sin\left(\tan^{-1}\left(\sqrt{2}\right)\right)}r_{f,free} \tag{8}$$

For a parallel topology, the fibers only contract axially during muscle contraction. Therefore, the fibers in a parallel topology are always contraction-limited. The initial fiber length and the number of fibers are bound by the length and width of the bounding envelope, respectively. Based on the orientation of the fibers in the bounded space, the fibers can only be arranged along the width dimension of the bounding envelope ($n_l = 1$). Inequality (5) can be used to realize $n_w$, which is equivalent to the number of fibers that can fit within the bounding envelope $n$. $n$ must be a positive non-zero integer. For bipennate topology, the fiber length and the maximum number of fibers that can fit in the bounding box depend on the fiber behavior. Although the constraints can provide insight into the fiber parameters, $l_{f,i}$ and $n$ are coupled and will depend on the minimum fiber clearance required to enable the fibers to fully contract, rotate, or both. An initial fiber length can be estimated from the following relationship.

$$l_{f,i} = \max\left(\frac{L - \max\left(r_f \sin(\beta)\right) - r_f \sin(\beta)}{\frac{\cos(\alpha)}{\cos(\alpha_i)}\cos(\beta)}\right) \tag{9}$$

This relationship depends on the minimum fiber clearance along the length dimension of the bounding envelope from the top-most pair of fibers, $\max\left(r_f \sin(\beta)\right)$. Assuming that at least one pair of fibers can fit along the width dimension of the bounding box (i.e., $n_w \geq 2$), this estimated initial fiber length must satisfy inequality (5). If this estimated initial fiber length satisfies the constraint (5), then this estimated initial fiber length is valid. However, this does not automatically indicate that a single pair of fibers is the maximum number of fibers that can fit along the width dimension of the bounding box. The appropriate $n_w$ and $n_l$ will need to be determined with this set of valid fiber dimensions based on the fiber boundary conditions. For the fiber contact boundary condition, the exact $n_w$ and $n_l$ only need to satisfy inequalities (5) and (6) respectively. For the pinned boundary condition, minimum clearance is needed to determine the fiber pin locations, such that the fibers do not interfere with one another during contraction and the resulting radial expansion. In addition, the $n_w$ and $n_l$ need to satisfy the following inequalities as well as inequalities (5) and (6) respectively, to ensure that the fibers remain inside the prescribed bounding envelope during muscle contraction:

$$\left(\frac{n_w}{2} - 1\right)\max\left(\frac{2r_f}{\cos(\beta)}\right) + 2r_f \cos(\beta) + l_f \sin(\beta) \leq \frac{W}{2} \tag{10}$$

$$\left(\frac{n_l}{2} - 1\right)\max\left(\frac{2r_f}{\sin(\beta)}\right) + 2r_f \sin(\beta) + l_f \cos(\beta) \leq L \tag{11}$$

The minimum clearance required between fibers along the width dimension of the bounding envelope is expressed as $\max\left(\frac{2r_f}{\cos(\beta)}\right)$ and the minimum clearance needed between fibers along the length dimension of the bounding envelope is computed as $\max\left(\frac{2r_f}{\sin(\beta)}\right)$. If this estimated initial fiber length violates inequality (3), the estimated initial fiber length must be recomputed with $n_w = 2$. Note that both $n_w$ and $n_l$ must be even and positive non-zero integers. The fibers are laterally arranged if $n_w \geq n_l$ and centrally arranged if $n_w < n_l$. These fiber arrangements are illustrated in Figure 4.

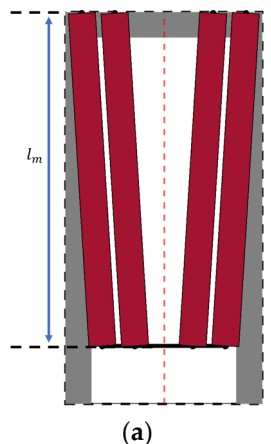 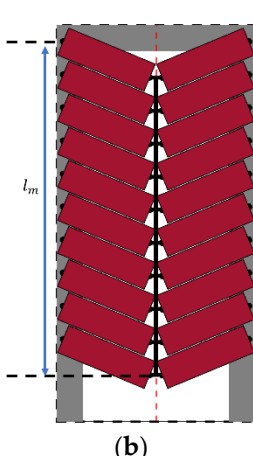 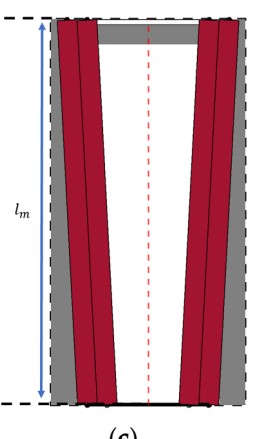 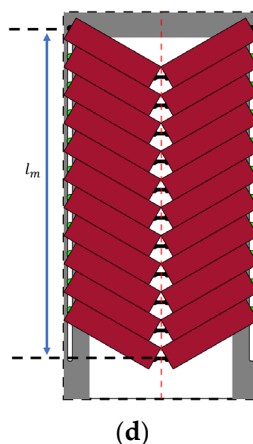

| (**a**) | (**b**) | (**c**) | (**d**) |

**Figure 4.** Muscle length for bipennate muscle bundles with (**a**) laterally arranged fibers and pinned boundary condition, (**b**) centrally arranged fibers and pinned boundary condition, (**c**) laterally arranged fibers and fiber contact boundary condition, and (**d**) centrally arranged fibers with fiber contact boundary condition.

## 3. Effects of Pennation Angle on Bundle Actuator Performance

### 3.1. Muscle Force-Strain Behavior

In order to understand the implications of varying fiber pennation angles on actuation performance, the muscle and fiber behavior during actuation is evaluated for a range of initial fiber pennation angles with the two fiber boundary conditions. Table 1 shows the

initial braid angle of the mesh sleeve for each fiber, as well as the prescribed bounding box parameter dimensions used throughout the study.

**Table 1.** System parameters.

| Parameter | Variable | Value |
|---|---|---|
| Initial braid angle | $\alpha_i$ | 30° |
| Bounding length | $L$ | 30.48 cm (12 in) |
| Bounding width | $W$ | 15.24 cm (6 in) |
| Bounding depth | $D$ | 2.54 cm (1 in) |

Muscle force $F_m$ is derived from the nonlinear force-strain relationship represented by the ideal virtual work model to apply for pennate configurations, by accounting for the component of force exerting in the direction of motion [13,15,19,20].

$$F_m = n\pi r_{f,i}^2 P\left( a\left(1-\varepsilon_f\right)^2 - b \right)\cos(\beta)$$
$$\beta = \sin^{-1}\left(\frac{\sin(\beta_i)}{\varepsilon_f}\right) \tag{12}$$
$$a = \frac{3}{\tan(\alpha_i)^2} \quad b = \frac{1}{\sin(\alpha_i)^2}$$

where $P$ is the pressure supplied to each fiber, and $a$ and $b$ are constants related to the initial braid angle. Figure 5 illustrates muscle force with respect to muscle strain at a constant applied pressure of 345 kPa (50 psi) for different bipennate muscle topologies and fiber constraints. We note that, for some initial pennation angles with the fiber contact boundary condition, the muscle strain becomes negative, indicating the extension of the muscle, or experiences a direction reversal, where it initially extends then contracts during actuation. For the pinned boundary condition, the muscle strain is always contractile, but the maximum strain depends strongly on the initial pennation angle.

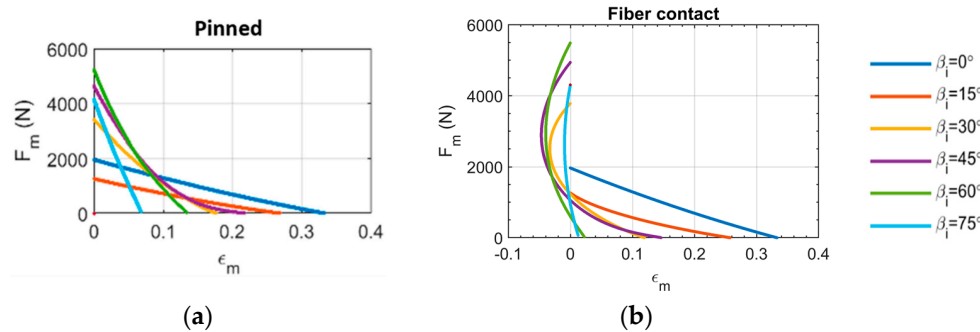

(**a**)  (**b**)

**Figure 5.** Muscle force–muscle strain behavior of parallel ($\beta_i = 0°$) and bipennate muscle bundles with pinned boundary condition (**a**) and fiber contact boundary condition (**b**). The legend shown on the far right indicates the fiber pennation angle of the muscle topology.

To understand this behavior, we consider the relationship between muscle strain $\varepsilon_m$ and fiber strain $\varepsilon_f$, where

$$\varepsilon_m = \frac{\Delta l_m}{l_{m,i}}$$
$$\varepsilon_f = \frac{\Delta l_f}{l_{f,i}} \tag{13}$$

The muscle length $l_m$ is defined as the distance, in the direction of muscle displacement, between the outer attachment point for the first pair of fibers and the inner attachment point of the last pair for fibers. Note that the center-to-center spacing between fibers in the fiber contact boundary condition changes with respect to fiber radial expansion and axial contraction, whereas it remains fixed in the pinned boundary condition. This is depicted

in Figure 5 for pinned fibers and fibers in contact. These variations in muscle length are expressed as

$$l_m = \begin{cases} l_f & \text{for parallel fibers} \\ l_f \cos(\beta) & \text{for bipennate lateral fibers} \\ \left(\frac{n}{2} - 1\right)\frac{2r_f}{\sin(\beta)} + l_f \cos(\beta) & \text{for bipennate central fibers} \end{cases} \tag{14}$$

Muscle displacement $\Delta l_m$ is measured as $\Delta l_m = l_{m,i} - l_m$ where $l_{m,i}$ is the initial muscle length. The muscle strain increases at the same rate as fiber strain increases for the parallel muscle topology, due to the fibers running parallel to the axis of muscle motion. It should be noted that a bipennate muscle with an initial pennation angle of $90°$ is incapable of muscle contraction along the direction of muscle motion.

Figure 6 plots muscle strain as a function of fiber strain. It shows that in rotation-limited topologies ($\beta_i \geq 41.86°$), as the initial pennation angle increases, the fiber strain decreases due to a decreasing range of fiber rotation. Hence, as the initial pennation angle increases for bipennate muscle topologies with the fiber contact boundary condition, the fiber strain increases at a faster rate than muscle strain. Furthermore, the negative (extensile) muscle strains exhibited by the fiber contact boundary condition with sufficiently large $\beta_i$ indicate that the contribution of the fiber radial expansion along the length dimension of the bounding envelope dominates the contribution of the fiber axial contraction. See Supplemental Video S1 for a representative animation of bipennate topology muscle actuation with fiber contact boundary conditions.

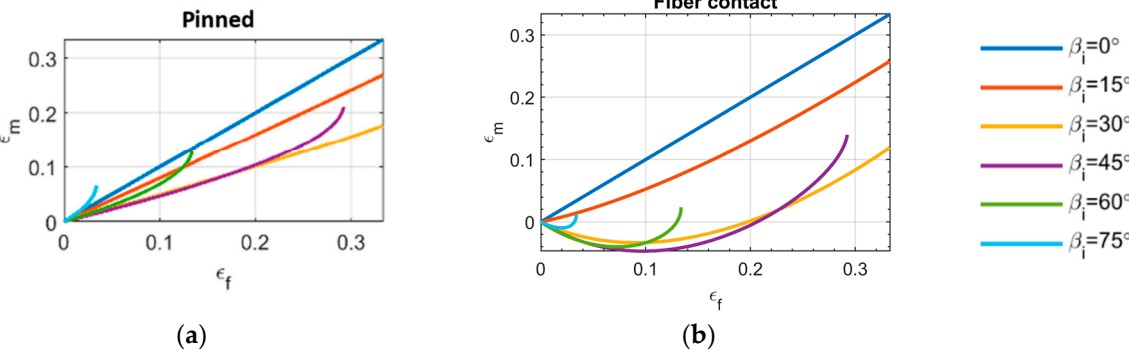

**Figure 6.** Muscle strain–fiber strain behavior of parallel ($\beta_i = 0°$) and bipennate muscle bundles with pinned boundary condition (**a**) and fiber contact boundary condition (**b**). The legend shown on the far right indicates the fiber pennation angle of the muscle topology.

Figure 7 dissects the muscle contraction behavior during actuation to show the contributions to overall muscle contraction from the radial expansion and axial contraction for a $\beta_i = 30°$ bundle with fiber contact boundary conditions. In the initial stage of actuation, the radial expansion dominates the axial contraction, such that the muscle experiences extensile motion. This elongation behavior will occur until the fiber axial contraction overcomes fiber radial contraction and drives the muscle to contract. However, depending on the muscle topology, the fiber axial contraction may never surpass the fiber radial expansion enough to enable muscle contraction. Similar effects have been reported in natural muscles; fiber pennation angles above approximately $55°$ have been shown to be incapable of producing muscle shortening in myomeres of fishes [21]. Biological muscles vary muscle thickness to impose the isovolumetric constraint on the muscle, while the artificial muscle bundle considered here keeps muscle thickness constant via the rigid frame that supports the outer muscle connections. The drawback of maintaining a constant muscle thickness is that it reduces the extent of fiber rotation, ultimately impacting muscle contraction. However, the pinned fiber boundary condition counters this drawback by eliminating the muscle elongation behavior by precisely locating the fibers, such that fiber contact is avoided, and

does not interfere with muscle motion. See Supplemental Video S2 for a representative animation of bipennate muscle actuation with pinned fiber boundary conditions.

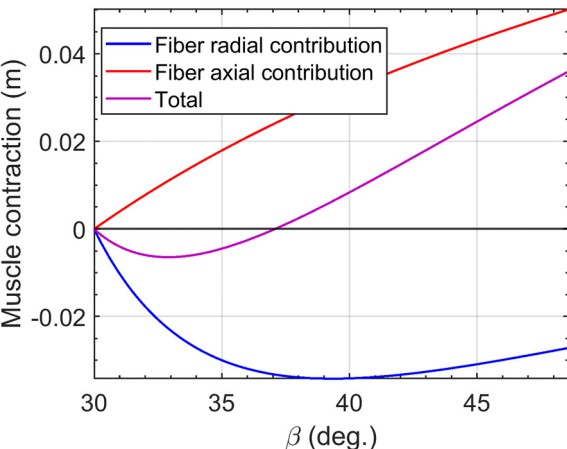

**Figure 7.** Variation in fiber radial and axial contributions to total muscle contraction with fiber pennation angle for a $\beta_i = 30°$ bundle with fiber contact boundary conditions.

The muscle force strain behavior for a bipennate muscle topology that experiences both extensile and contractile motion is broken down to illustrate the muscle displacement direction reversal in Figure 8. Figure 8a illustrates the muscle force during actuation for a bipennate muscle of an initial pennation angle of 45° under the fiber contact constraint. A constant pressure of 345 kPa (50 psi) is supplied to all the fibers. The series of images in Figure 8b provide visual snapshots of the muscle state at different stages of muscle actuation. The distance between the upper and lower blue lines is the muscle length at the specified actuation. The figure illustrates that, as the muscle contracts from the initial (blocked force) condition, the fiber radial expansion dominates the muscle movement and the muscle begins to extend while the muscle force decreases. Once the fiber axial contraction overcomes the fiber radial expansion, the muscle transitions to contractile motion. Thus, the muscle can experience the same muscle strain at different stages of fiber contraction, as can be seen at point (3), where the muscle topology appears noticeably dissimilar to point (2), even though $\varepsilon_m = -0.025$ in both cases. Depending on the muscle topology, the contractile motion may be capable of bringing the muscle back to the original muscle length, as seen at point (4) and transition to muscle contraction as seen in point (5). However, for some other bipennate muscle topologies with the fiber contact constraint applied, the fiber axial contraction is incapable of overcoming the fiber radial expansion. This results in the muscle only experiencing extensile motion.

Analogous to architectural gear ratio (AGR) in biomechanics studies, the transmission ratio can be used to relate the fiber behavior to the muscle behavior. The transmission ratio $TR$ analyzed in this study is defined as the ratio of change in muscle length to the change in fiber length, as shown in Equation (15).

$$TR = \frac{\Delta l_m}{\Delta l_f} \tag{15}$$

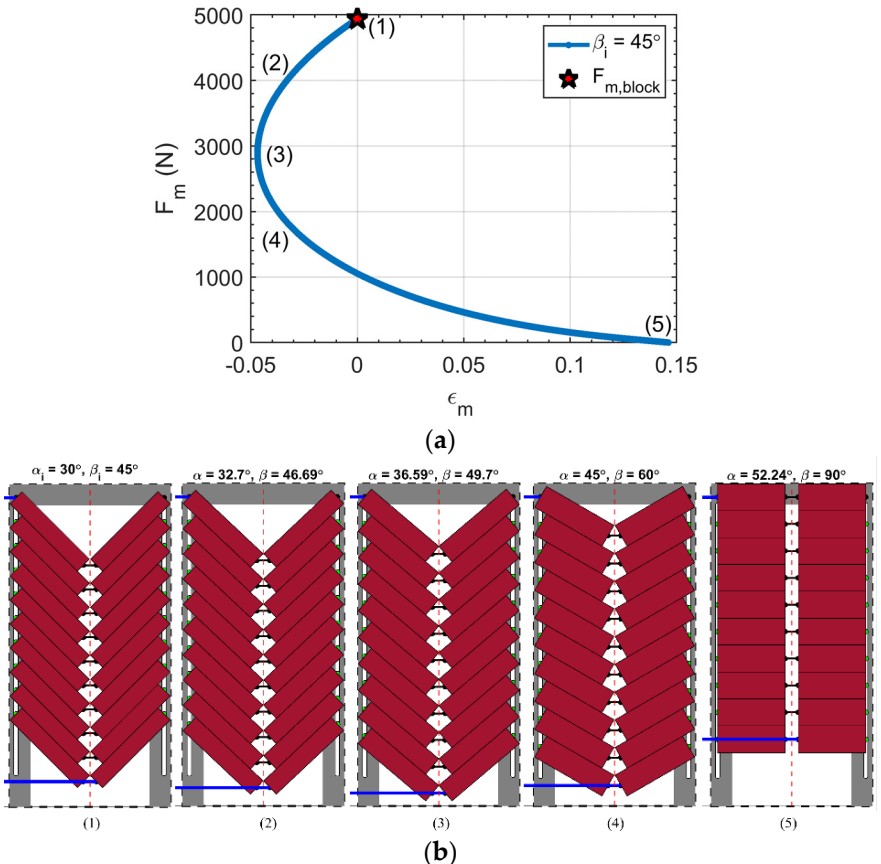

**(a)**

**(b)**

**Figure 8.** (**a**) Muscle force–muscle strain behavior of bipennate muscle bundle with $\beta_i = 45°$ and fiber contact boundary condition. The numbered label on the plot indicates the sequential change in muscle force and muscle strain during fiber actuation. (**b**) Visual representation of bipennate muscle bundle motion with fiber contact boundary condition. Each numbered image corresponds to different stages in fiber actuation and aligns with the numbered labels in (A). (1) is the initial or blocked force condition, (2) is $\varepsilon_m = -0.025$ during extensile motion, (3) is $\varepsilon_m = -0.025$ during contractile motion, (4) is when the muscle returns to its initial muscle length, and (5) is the free muscle contraction condition.

Figure 9 shows how the transmission ratio varies with respect to fiber pennation angle for different fiber boundary conditions. In the parallel topology, the muscle contraction rate and fiber contraction rate are equivalent (i.e., $TR = 1$), because no fiber rotation occurs. The transmission ratio for the parallel muscle topology is shown as a red star marker. A transmission ratio greater than one indicates that the muscle topology amplifies fiber contraction; a ratio less than one indicates that topology reduces contraction. Figure 9 shows that, for all bipennate muscles, the transmission ratio increases as the pennation angle increases. The rates of change of transmission ratio with pennation angle observed here differ from that previously reported for bipennate bundle actuators with fixed fiber geometries, due to the spatial constraint on muscle envelope and the corresponding fiber sizing considered here [13,22]. The pinned fiber boundary condition analysis shows that all bipennate muscle topologies considered producing transmission ratios greater than 1. For bipennate muscles with fibers in contact, both positive and negative transmission ratios are observed, and ratios are generally less than 1. The negative transmission ratio values align with the elongation behavior observed in bipennate muscles, with the fiber contact boundary condition that was seen in Figures 5b and 6b. Knowledge of bipennate variable gearing and muscle force-strain space gives insight into matching muscle output demands of speed or force for a broad range of movements. Although some have explored the load dependences of muscle-like soft actuators with unipennate structures, their study enables

the spatial bound of the actuator to change. Thus, their results were only able to isolate the dependence of the pennation angle and number of contractile units on the architectural gear ratio (AGR) [23]. The equal spatial envelope constraint imposed in the analysis presented in this paper allows for a fair and direct comparison between configurations, as well as providing practical insights into which pennation configuration is best to accomplish their design goals, and what tradeoffs exist when using a pennate configuration, to optimize a certain actuation characteristic at the expense of others.

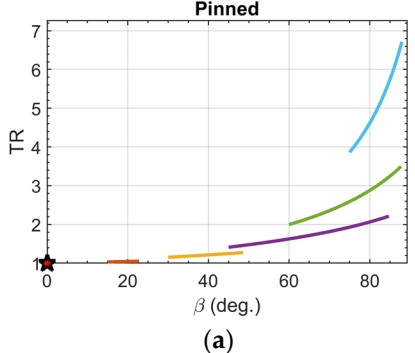
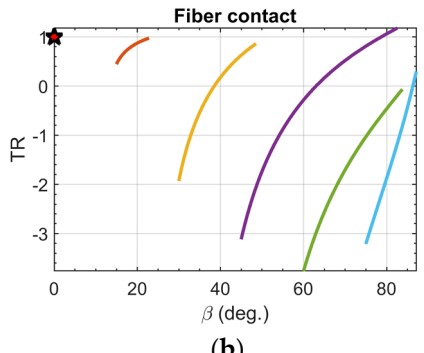
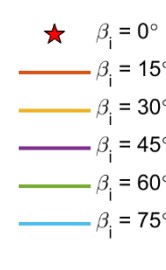

**Figure 9.** Variation in muscle transmission ratio of parallel ($\beta_i = 0°$) and bipennate muscle bundles with pinned boundary condition (**a**) and fiber contact boundary condition (**b**). The legend shown on the far right indicates the fiber pennation angle of the muscle topology.

### 3.2. Peak Muscle Displacement

The peak muscle displacement reaches a maximum and then decreases significantly as the initial fiber pennation angle increases, as shown in Figure 10. However, this relationship is nonlinear due to the extent of fiber contraction and rotation behavior in the muscle bundle. At small initial pennation angles, the fibers are arranged laterally where the fiber length makes a larger contribution to the muscle length. The fiber lengths at small initial pennation angles for bipennate muscles are almost equal to the fiber lengths of the parallel muscle, as shown in Figure 11. This is due to the fibers being contraction-limited and the length dimension of the bounding envelope determining the initial fiber length. Fibers at these small initial pennation angles rotate with fiber contraction and assist in muscle contraction, such that the peak muscle displacement exceeds that of the parallel topology. As the initial pennation angle increases, the initial fiber length is increasingly driven by the width dimension of the bounding envelope and thus decreases. This is even more so for rotation-limited fibers, since the fibers need to fully rotate.

One may intuitively assume that a bipennate configuration, in which the fibers simultaneously fully contract and fully rotate, should result in a greater muscle displacement, as compared to a parallel topology. This was observed in a previous study that considered fibers of prescribed length, with varying initial pennation angles [13]. Our analysis of the prescribed bounding envelope case considered here partially supports this intuition, as the peak muscle displacement of bipennate muscle topologies approaches a local maximum at the fiber contraction-rotation limited boundary. However, this local maximum is noticeably less than the peak muscle displacement of the parallel topology, since the combination of fiber sizing, rotation, and boundary conditions dictates the extent of muscle contraction. This is a direct implication of the prescribed bounding volume and a design objective of effectively utilizing the available space. The small jumps in the peak muscle displacement are associated with the changes in the number of fibers in the muscle bundle. We also note that, for a range of bipennate muscle topologies with the fiber contact boundary condition, the peak muscle displacement falls below zero, indicating that those configurations have a larger magnitude of extension than compression, resulting in a peak muscle displacement in the extensile direction.

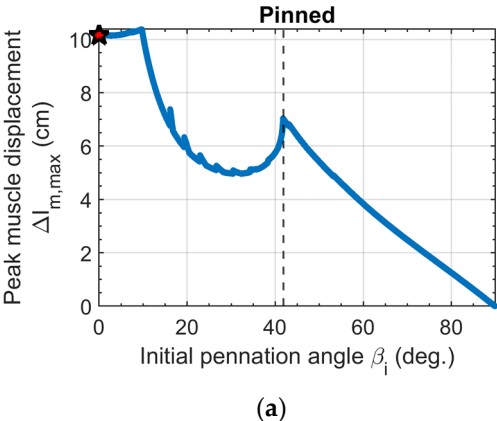

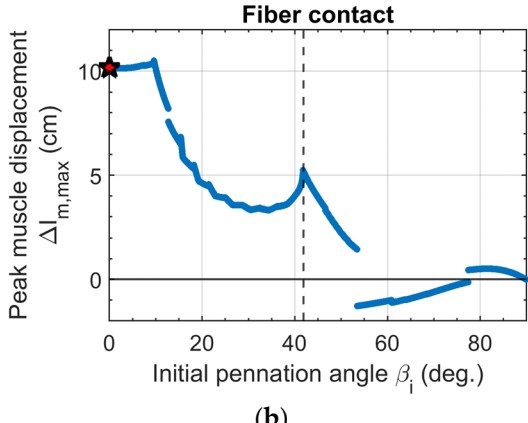

(**a**)                                                           (**b**)

**Figure 10.** Variation in peak muscle displacement for muscle bundle design with (**a**) pinned boundary condition and (**b**) fiber contact boundary condition. The red star indicates a parallel ($\beta_i = 0°$) muscle bundle topology, and dashed vertical line indicates the boundary between fiber contraction-limited (left of the line) and fiber rotation-limited (right of line) topologies.

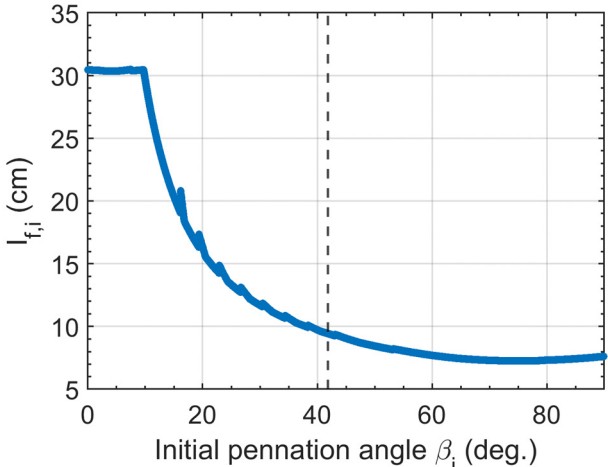

**Figure 11.** Variation in initial fiber length with initial pennation angle with the pinned boundary condition. The dashed vertical line indicates the boundary between fiber contraction-limited (left of the line) and fiber rotation-limited (right of line) topologies.

### 3.3. Muscle Blocked Force

The muscle blocked force (i.e., the force at zero strain) varies nonlinearly with the initial pennation angle. This condition represents the maximum force generation of the muscle, and can be expressed as a function of applied pressure and initial pennation angle as

$$F_{m,block} = F_m\big|_{\varepsilon_f=0} = n\pi r_{f,i}^2 P(a-b)\cos(\beta_i) \tag{16}$$

The muscle blocked force is generally expected to increase with increasing initial fiber pennation angle, since a larger number of fibers can be packed in the muscle bundle. However, this is not always the case for spatially bound muscle bundles, as seen in Figure 12, due to the fiber arrangement. The lateral or central arrangement of the fibers depends on the number of fibers that can fit along the width of the bounding envelope $n_w$ and number of fibers that can fit along the length of the bounding envelope $n_l$. Figure 13 illustrates the range of initial pennation angles, where fibers are laterally and centrally arranged. The fiber contact boundary condition shows a similar trend in the number of fibers, but the lack of clearance space between the fibers permits additional fibers to fit in the bounding envelope at certain initial pennation angle values.

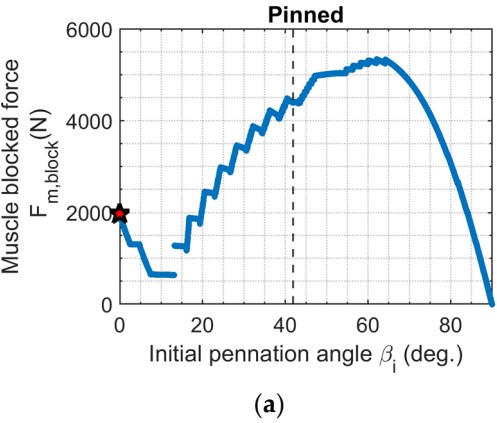
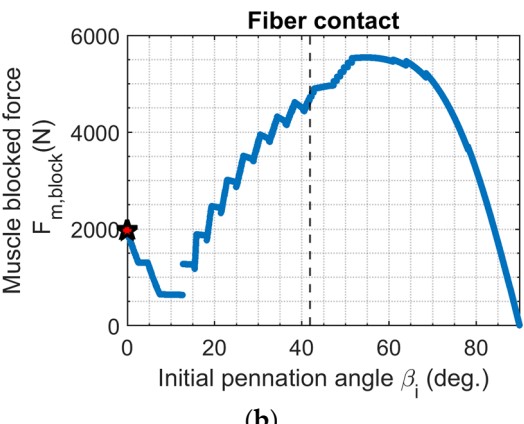

**Figure 12.** Variation in muscle blocked force at a constant applied pressure of 345 kPa (50 psi) with initial pennation angle for (**a**) pinned boundary condition and (**b**) fiber contact boundary condition. The red star indicates a parallel ($\beta_i = 0°$) topology, and a dashed vertical line indicates the boundary between fiber contraction-limited (left of the line) and fiber rotation-limited (right of line) topologies.

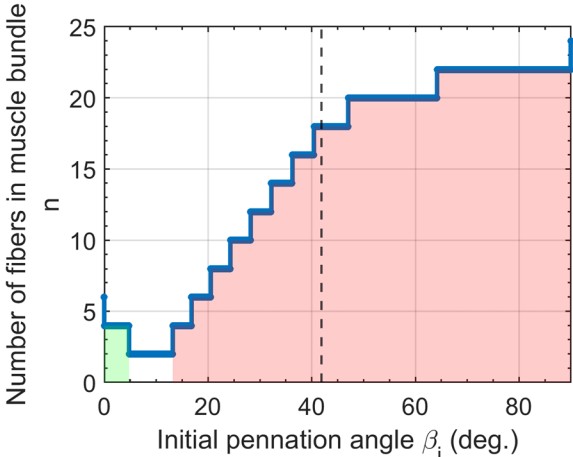

**Figure 13.** Variation in the number of fibers in muscle bundle with initial pennation angle with the pinned boundary condition. The green shaded region is associated with laterally arranged fibers and the pink shaded region indicates centrally arranged fibers. The white region indicates where only one pair of fibers can fit in the prescribed bounding envelope.

The fiber diameter also plays a part in fiber arrangement and force capacity. The initial fiber radius depends on the contraction–rotation behavior, such that contraction-limited fibers have an appropriate initial fiber radius to enable full radial expansion, while rotation-limited fibers can have larger radii depending on the extent of radial expansion, as shown in Figure 14.

The fiber radius and length do not vary with respect to fiber boundary conditions. Thus, for a given initial pennation angle, the muscle bundle blocked force profile with the fiber contact condition is equal to (in cases where the same number of fibers are present) or slightly larger than (in cases where additional fibers are present) that with the pinned condition, as shown in Figure 12. The jumps in the blocked force profiles correspond to variation in the number of fibers in the muscle bundle. This analysis also shows that a large range of bipennate muscle topologies can achieve a muscle-blocked force significantly greater than that of the parallel muscle topology, with the largest blocked force configuration, exhibiting more than 2.5 times that of the parallel topology. Although the fiber clearance required by the pinned condition decreases the number of fibers in the muscle bundle compared to the fiber contact condition, the pinned condition is generally comparable to that of the fiber contact condition.

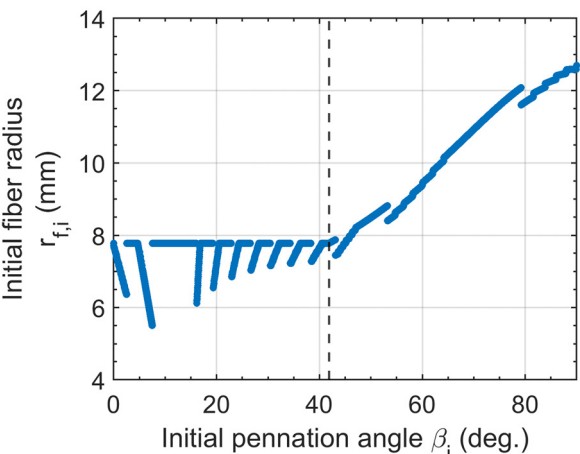

**Figure 14.** Variation in initial fiber radius with initial pennation angle with the pinned boundary condition. The dashed vertical line indicates the boundary between fiber contraction-limited (left of the line) and fiber rotation-limited (right of line) topologies.

### 3.4. Muscle Stiffness

In understanding the muscle force–strain relationship, the muscle displacement direction reversal suggests that bipennate muscle topologies can be designed as both positive and/or negative stiffness structures. In this analysis, the following formulation is used to compute the muscle stiffness $k_m$ behavior during actuation.

$$k_m = \frac{dF_m}{d\Delta l_m} = \frac{\frac{\partial F_m}{\partial \alpha}}{\frac{\partial \Delta l_m}{\partial \alpha}} \tag{17}$$

The applied pressure is held constant at 345 kPa (50 psi) to observe the stiffness during isobaric contraction. Figure 15 shows that operating stiffness decreases with increasing pennation angle for muscle topologies with pinned fiber boundary condition, but stiffness always remains positive. Negative stiffness behavior is observed in muscle topologies with fibers in contact, as the muscle length increases with decreasing force. Figure 16 illustrates a bipennate muscle with fiber contact that transitions from extensile to contractile motion. The stiffness approaches an asymptote or an instance of infinite stiffness when the muscle transitions from extensile to contractile motion. This infinite stiffness occurs at a mechanical singularity where the fiber radial expansion and axial contraction can no longer allow the muscle to extend without violating the fiber contact boundary condition constraint. This mechanical singularity prevents a conclusive statement being made on the operating stiffness range for muscle topologies with the fiber contact boundary condition. On the other hand, the operating stiffness range for muscle topologies with pinned fibers increases with muscle topologies approaching the fiber contraction and rotation-limited boundary, where the fibers in the muscle can fully contract and fully rotate.

### 3.5. Isobaric Work Output

Although a clear tradeoff is observed between muscle output force and muscle displacement, the isobaric work provides a more complete understanding of muscle output capacity. Isobaric work output is evaluated by understanding the muscle force behavior during actuation at a constant pressure. An equivalent pressure of 345 kPa (50 psi) is supplied in this analysis to observe an impartial comparison.

$$W_{isobaric} = \int_{l_{m,i}}^{l_{f,free}} F_m dl_m \tag{18}$$

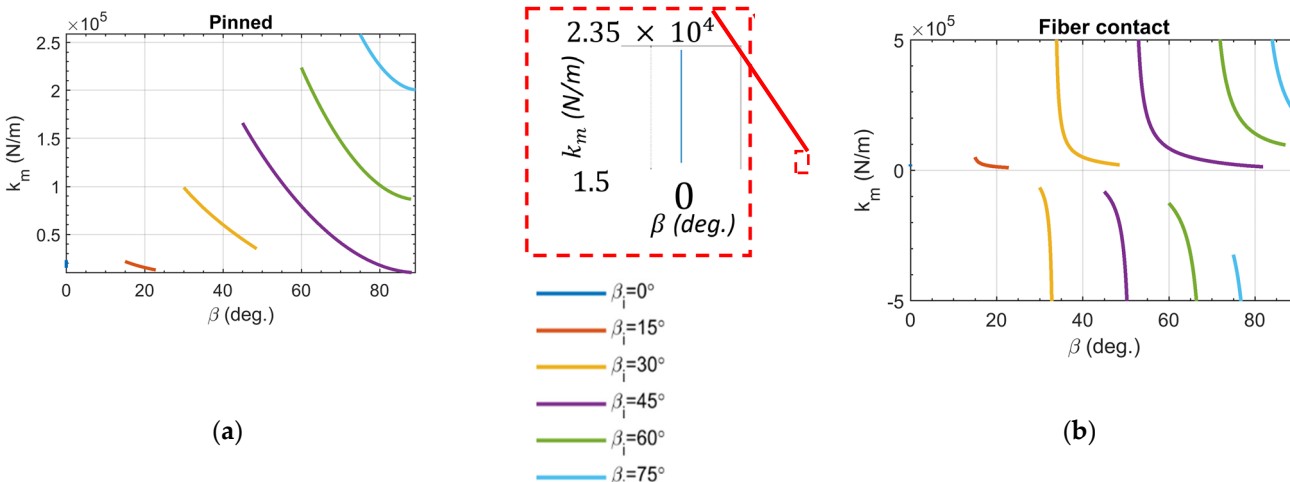

**Figure 15.** Variation muscle stiffness behavior of parallel ($\beta_i = 0°$) and bipennate muscle bundles with pinned boundary condition (**a**) and fiber contact boundary condition (**b**). The legend shown on the far right indicates the fiber pennation angle of the muscle topology. The inset plot shown in the red dashed box illustrates the muscle stiffness range for a parallel muscle bundle. The muscle stiffness range for the parallel topology is the same for both boundary conditions.

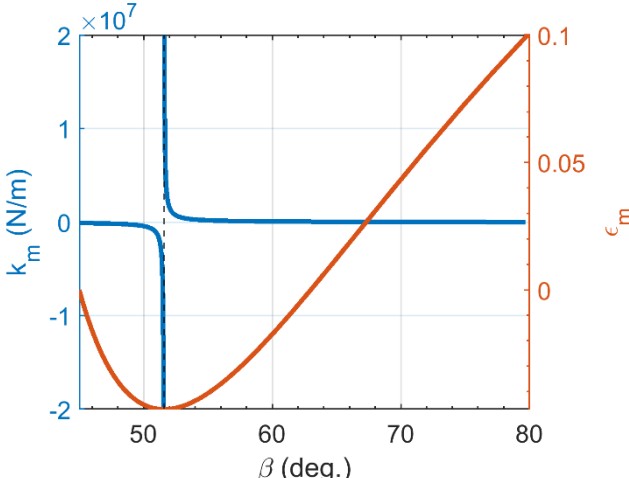

**Figure 16.** Realized negative muscle stiffness behavior for bipennate muscle bundles with fiber contact boundary condition. The blue profile illustrates a variation in muscle stiffness during fiber actuation, and the orange profile corresponds to a change in muscle strain during fiber actuation.

Isobaric work depends on the muscle output force, so it is expected for the isobaric work to resemble that of the muscle blocked force, as seen in Figure 14. However, this is not true for bipennate muscle topologies with the fiber contact boundary condition shown in Figure 17b. The negative work corresponds to the muscle topologies that experience extensile motion. The peak muscle displacement analysis in Figure 10b shows a smaller range of bipennate muscle topologies with the fiber contact boundary condition, where the extensile motion exceeds that of the contractile motion, while the isobaric work output capacity shows a larger range of bipennate muscle topologies; the elongation motion does not contribute to positive isobaric work. The jumps in the profile are associated with a change in the number of fibers in the muscle bundle. Since elongation behavior does not exist for muscle topologies with the pinned boundary condition, the isobaric work is always positive, as seen in Figure 17a. Furthermore, it shows that a bipennate muscle topology with the pinned boundary condition can be designed with an isobaric work output capacity larger than that of the parallel muscle topology of the same size.

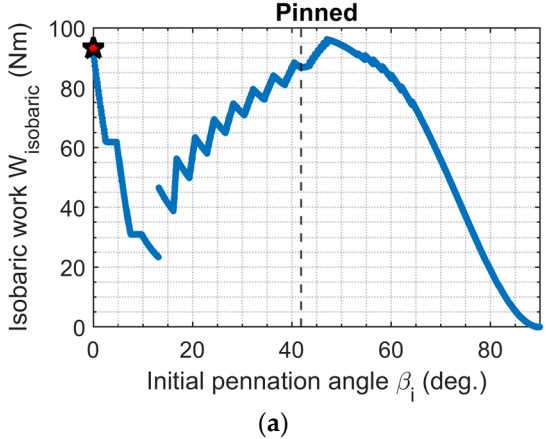
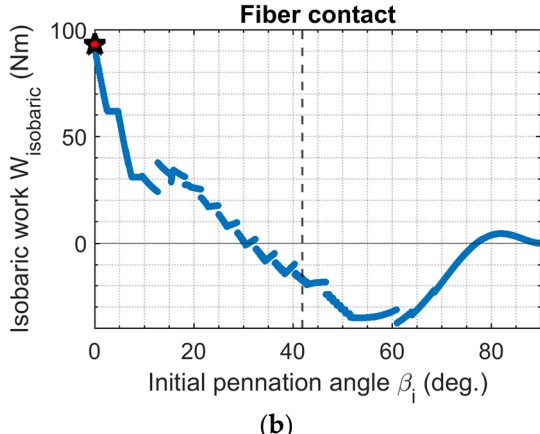

(a)  (b)

**Figure 17.** Variation in isobaric work output capacity with initial pennation angle for (**a**) pinned boundary condition and (**b**) fiber contact boundary condition. The red star indicates a parallel ($\beta_i = 0°$) muscle bundle topology, and dashed vertical line indicates the boundary between fiber contraction-limited (left of the line) and fiber rotation-limited (right of line) topologies.

### 3.6. Isotonic Work Output

Isotonic work output is evaluated by understanding the muscle behavior during muscle actuation when a constant load $F_{load}$ is applied. The following expression is used to compute isotonic work.

$$W_{isotonic} = F_{load} \int_{l_{m,i}}^{l_{m,free}} dl_m \tag{19}$$

Figure 18 shows that isotonic work scales linearly with the applied load for a given $\beta_i$. The isotonic work profile in Figure 18 resembles the peak muscle displacement seen in Figure 10, since isotonic work depends on the muscle displacement behavior of the muscle. There are bipennate muscle topologies capable of isotonic work that slightly exceed that of the parallel muscle topology with either boundary condition. Similar to the isobaric work of muscle topologies with the fiber contact boundary condition, the negative isotonic work observed in bipennate muscle topologies with the fiber contact boundary condition is a result of the extension behavior.

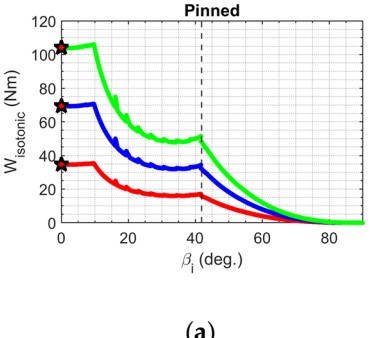
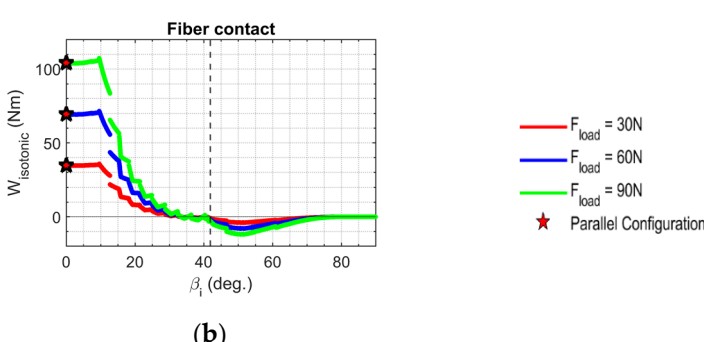

(a)  (b)

**Figure 18.** Variation in isotonic work output capacity of parallel ($\beta_i = 0°$) muscle bundles with pinned boundary condition (**a**) and fiber contact boundary condition (**b**). The legend on the far right indicates the fiber pennation angle of the muscle topology..

## 4. Conclusions

This paper establishes a method of designing bipennate fluidic artificial muscle bundle topologies under spatial bounding volume constraints. The muscle bundles were parameterized for effectively utilizing the prescribed bounding envelope. Two fiber boundary conditions—fiber contact and pinned—were explored to investigate the implications of incorporating an idealized bio-inspired connective tissue functionality, as compared to

maintaining clearance between fibers via fixed pin joints, respectively. This is the first study to quantify the effects of pennation angle on the force, contraction, stiffness, and work output of fluidic artificial muscle bundles, while maintaining a uniform spatial envelope for the muscle bundle. In evaluating the bundle geometry required to maintain the same spatial envelope for each pennation angle configuration, a fair and direct comparison could be established between configurations, including parallel and pennate muscle bundles. The analysis revealed that, for the given spatial volume constraint, bipennate topologies can be found to produce muscle output force, muscle contraction, muscle stiffness, or work output capacity larger than that of a parallel muscle bundle topology. To our knowledge, no previous study has compared parallel and pennate configurations, or various pennate configurations, under this equal spatial envelope constraint. This spatial envelope constraint is believed to be important for various practical actuator design applications that are limited to a given volume envelope to house the actuator.

This study was also the first to investigate the fiber contact boundary condition for the fluidic artificial muscles that act as the 'fibers' within the pennate bundle, and compare the implications of this boundary condition to the more commonly studied pinned boundary condition [13,15]. The fiber contact boundary condition is of interest, as it is more closely bio-inspired, and can produce extensile motion as well as negative stiffness behaviors in some configurations. Thus, depending on the fiber boundary conditions, a bipennate bundle may offer distinct advantages over a more conventional parallel bundle.

The fiber contact boundary condition enables additional fibers to be packed into the fixed bounding envelope through the elimination of minimum fiber clearance required in the pinned boundary condition. However, due to decreased contraction, fiber contact configurations produce less work output than pinned configurations over most of the range of initial pennation angles. Contrary to intuition, configuring a bipennate topology to simultaneously maximize fiber contraction and fiber rotation does not yield the largest muscle stroke, and results in less contraction than that of a parallel topology with an equal bounding envelope. Rather, peak muscle displacement is produced by a bipennate topology with a small (but non-zero) initial pennation angle that exploits both longer fibers and rotation effects. In addition, an analysis of muscle behavior found that the competing effects of fiber axial contraction and radial expansion in the fiber contact boundary condition determine the overall muscle motion, such that a bipennate topology can exhibit extensile motion. The extensile motion found in muscle topologies with the fiber contact bounding condition also enables the bipennate muscle topology to behave as a negative stiffness structure. Furthermore, the mechanical singularity was discovered as a byproduct of the fiber contact boundary condition.

Future work will include experiments to investigate these trends in bipennate muscle bundles with the fiber contact boundary condition. To advance our understanding in the design of pennate muscle topologies, future work should explore implications of muscle topology design on efficiency in tracking a dynamic motion, and evaluate muscle performance sensitivity to variation in muscle thickness during actuation.

**Supplementary Materials:** The following supporting information can be downloaded at: https://www.mdpi.com/article/10.3390/act11030082/s1, Video S1: fibercontact_beta_i_60, Video S2: pinned_beta_i_60. Supplementary videos are provided as representative illustrations of muscle bundles during actuation. Video S1 corresponds to a muscle bundle with fiber contact boundary condition while Video S2 corresponds to a muscle bundle with pinned boundary condition.

**Author Contributions:** Conceptualization, E.D. and M.B.; methodology, E.D.; software, E.D.; validation, E.D. and M.B.; formal analysis, E.D.; investigation, E.D.; resources, M.B.; data curation, E.D.; writing—original draft preparation, E.D.; writing—review and editing, E.D. and M.B.; visualization, E.D. and M.B.; supervision, M.B.; project administration, M.B.; funding acquisition, M.B. All authors have read and agreed to the published version of the manuscript.

**Funding:** This research was funded by the National Science Foundation, grant number 1845203.

**Institutional Review Board Statement:** Not applicable.

**Informed Consent Statement:** Not applicable.

**Data Availability Statement:** Data are contained within the article.

**Acknowledgments:** This work was supported by the Faculty Early Career Development Program (CAREER) of the National Science Foundation under NSF Award Number 1845203 and Program Manager Irina Dolinskaya. An NC Space Grant Graduate Research Fellowship also supported this research.

**Conflicts of Interest:** The authors declare no conflict of interest. The funders had no role in the design of the study; in the collection, analyses, or interpretation of data; in the writing of the manuscript, or in the decision to publish the results.

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
