# Peer review of "Implications of Spatially Constrained Bipennate Topology on Fluidic Artificial Muscle Bundle Actuation"

_actuators, doi:10.3390/act11030082_

Round 1

Reviewer 1 Report

Comments to authors:

1- The novelty is not clear.

2- Page 3, section 2.2, the authors refere to the figure 2.b before 2.a. Please correct their order.

3- Page 4, section 2.3 is not clear. please explain what is the diffrent between the configrations in figure 3 and 4?.

4- The captions of figures 3 and 4 are similar. Please either merge them or rewrite them to show the diffrent. Please mention to the sub figures a, b , and c seperetly. 

5- What is the references for equations 2 and 10? 

6- Is the system constructed practically? 

7- Please extend the litereture review and compare the proposed system. 

Reviewer 2 Report

In this manuscript, the authors investigate the design of pennate topology fluidic artificial muscle bundles under spatial constraints. Two fiber boundary conditions – fiber contact and pinned – were explored to investigate the implications of incorporating an idealized bioinspired connective tissue functionality as compared to maintaining clearance between fibers via fixed pin joints, respectively. For the acceptance of this manuscript, the authors may solve the following issues

  1. It will be better if the authors could combine the theoretical results with the FEA simulation, so the whole analysis procedure will be more convincing.

  1. There are some writing errors, for example, in Page 2 Line 73, the “establash” should be “establish”. Please check the manuscript carefully.

Reviewer 3 Report

The paper discusses the spatially constrained bipennate topology on fluidic artificial muscle bundle actuation. The bipennate structure shows a compared mechanical performance similar to the parallel topology. The paper claims that the bipennate topology reveals insights between the fibers and the muscle behaviours. However, a proper comparison needs to be reported.

The idea of the pennation angle has been addressed previously in textile-based soft actuators. Please cite [1,2].

[1] Mechanical Programming of Soft Actuators by Varying Fiber Angle, Soft Robotics, 2015.

[2] High-Performance Perpendicularly-Enfolded-Textile Actuators for Soft Wearable Robots: Design and Realization, IEEE, 2020.

English needs extensive revision. For example:

In the abstract:

Examination of natural muscles have shown -> Examination of natural muscles has shown

This study therefore enables tailoring -> This study, therefore, enables tailoring

Introduction:

is comprosed -> is composed

Analagously -> Analogously

The primary contributions of this paper are to establash -> establish

If the mucle bundle -> muscle

Section 2 presents the system formulation with method -> with a method

Section 3 discusses and examines effects of boundary -> the effects of

Round 2

Reviewer 1 Report

No further corrections are required. Q